# Spatio-Temporal Heterogeneity-Oriented Graph Convolutional Network for Urban Traffic Flow Prediction

**DOI:** 10.3390/s25165127

**Published:** 2025-08-18

**Authors:** Xuan Li, Muyang He, Dong Qin, Tianqing Zhou, Nan Jiang

**Affiliations:** 1School of Information and Software Engineering, East China Jiaotong University, Nanchang 330013, China; lixuan@ecjtu.edu.cn (X.L.); jiangnan1018@gmail.com (N.J.); 2School of Information Engineering, Nanchang University, Nanchang 330031, China

**Keywords:** VANET, graph convolutional network, traffic flow prediction, space heterogeneity, cross-domain data

## Abstract

In the realm of urban vehicular ad hoc networks (VANETs), cross-domain data has constituted a multifaceted amalgamation of information sources, which significantly enhances the accuracy and response speed of traffic prediction. However, the interplay between spatial and temporal heterogeneity will complicate the complexity of geographical locations or physical connections in the data normalization. Besides, the traffic pattern differences incurred by dynamic external factors also bring cumulative and sensitive impacts during the construction of the prediction model. In this work, we propose the spatio-temporal heterogeneity-oriented graph convolutional network (SHGCN) to tackle the above challenges. First, the SHGCN analytically employs spatial heterogeneity between urban streets rather than simple adjacency relationships to reveal the spatio-temporal correlations of traffic stream movement. Then, the air quality data is taken as external factors to identify the traffic forecasting trend at the street level. The hybrid model of the graph convolutional network (GCN) and gated recurrent unit (GRU) is designed to investigate cross-correlation characteristics. Finally, with the real-world urban datasets, experimental results demonstrate that the SHGCN achieves improvements, with the RMSE and MAE reductions ranging from 2.91% to 41.26% compared to baseline models. Ablation studies confirm that integrating air quality factors with traffic patterns enhances prediction performance at varying degrees, validating the method’s effectiveness in capturing the complex correlations among air pollutants, traffic flow dynamics, and road network topology.

## 1. Introduction

With the increasing popularization of urbanization, urban traffic flow forecasting has begun to attract widespread attention in smart cities [1,2]. Urban environments involve complex road network structures and external factors, e.g., weather, holidays, and road maintenance, thereby necessitating the rational utilization of transportation resources to acquire the massive spatio-temporal sequence information to attain accurate predictions [3]. As shown in Figure 1, the vehicular ad hoc networks (VANETs) can enable the comprehensive connectivity among vehicles, roads, people, and services, deemed as an efficient tool to boost vehicle intelligence and autonomous driving in next-generation communication networks [4,5,6]. In particular, IoVs allow for various collections and sharing of data, such as vehicle locations, speeds, and routes, thereby providing traffic managers with rich real-time data and increasing the precision of traffic prediction [7,8,9].

Seeking to sort out the intricate spatio-temporal relationship in traffic flow data, most existing works tend to utilize the derivatives of the convolutional neural network (CNN) or recurrent neural network (RNN) [10,11,12]. Despite the significant advances in applying neural networks to capture traffic flow features, the effectiveness is still limited in non-Euclidean spaces [13]. In [14,15], the graph convolutional network (GCN) is built to describe connectivity between different roads and associated traffic flows via mapping the road network topology into graphs, vertex weights, and edge weights. It is demonstrated that blending the physical with semantic data from the road network can offer a deeper insight into traffic dynamics, and the graph-based models exhibit that fluctuations in traffic flow are highly related to road topologies [16,17].

As shown in the traffic scenario of Figure 1, there are five streets, labeled from A to E. The traffic congestion occurs in street D, and the upstream of street D (i.e., streets A and B) and the downstream (i.e., street E) are affected due to the adjacency. Also note that the traffic of streets B and E still exhibits a notable correlation even without any connection, whereas the adjacent street C is uncongested. Also hinted at by Figure 1 is that existing works tend to merely consider the traffic relationship between adjacent streets, failing to describe the intricate correlations across different street blocks, especially those non-adjacent ones.

Moreover, it is acknowledged that the traffic volume is also related to regional features where roads are located [18,19]. In Figure 1, different regions, i.e., residential, industrial, or commercial, are typically characterized by distinct and varying traffic flows. Heieh et al. in [20] showed that the urban air quality is closely associated with traffic conditions in proximity. Nevertheless, such an illuminating relationship has not been explicitly considered in existing works regarding traffic flow forecasting. Actually, this type of spatial–temporal dynamical correlation between road networks, air quality, and regional traffic flows would pose the following two significant challenges for accurate predictions:The traffic flow in cities tends to follow a radial pattern along the road network, which is typically non-linear, implying the correlation of traffic flows between non-adjacent roads, but not between adjacent ones, e.g., between street C and D in Figure 1. Therefore, how to describe such uncertain correlations with the graph-based topology poses challenges.The correlation between air quality and traffic flow manifests in both spatial and temporal dimensions. A bidirectional interaction exists where vehicle emissions directly affect pollutant concentrations, while air quality changes (e.g., visibility) reciprocally influence traffic flow—particularly during rush hours. More importantly, different air pollution components (e.g., particulate matter [PM], oxides, and sulfides) exhibit strong regional characteristics. The superposition of these temporal, spatial, and compositional correlations intuitively leads to more complex and unstable predictions.

Inspired by the above challenges, we design a new traffic flow model, spatio-temporal heterogeneity-oriented graph convolutional network (SHGCN), to effectively capture the distinct traffic patterns stemming from the dynamic interactions between spatial heterogeneity and external factors. We first model the graph-based topological structure by characterizing the spatial heterogeneity between road segments and then propose a method of horizontally expanding the traffic feature matrix, facilitating the integration of air quality factors with enhanced road topologies. Finally, a convolution mechanism is used to fuse spatio-temporal correlations into the prediction across varying scales as well as pollutant components. The contributions are summarized as follows:The SHGCN is designed to capture dynamics and intricate relationships among versatile factors in the urban traffic flow by exploring the spatial–temporal heterogeneity between domains for the cross-mode traffic flow prediction, in which the correlations and dynamics among factors (e.g., urban road networks, air quality, and traffic data) are captured.Involving the space heterogeneity, a temporal encoder using temporal convolutions and gating mechanisms is devised, and an enhanced graph convolutional network is designed to explore the hierarchical correlations between traffic patterns and road topology, where the similarity clusters of heterogeneity degrees are deduced via the Bernoulli distribution.A *K*-means-based strategy is established to analyze the relationships between traffic flow and air composition by integrating the traffic with air quality features in order to enhance the stability of the hybrid prediction model which combines the gated recurrent unit (GRU) with the convolutional neural network (GCN).Extensive experiments are conducted with two real-world datasets, and numerous results are provided to demonstrate the performance improvement in the SHGCN over eight existing baseline methods, in terms of the root mean square error (RMSE), mean absolute error (MAE), and accuracy metrics.

The rest is organized below. Related works are given in Section 2. Section 3 elaborates on the details of the proposed SHGCN. In Section 4, experiment results are used to show the effectiveness of the proposed SHGCN. Lastly, Section 5 presents the conclusions and further investigation.

## 2. Related Works

In this section, related works on traffic flow prediction are listed from the perspectives of spatio-temporal and external features, respectively.

### 2.1. Traffic Flow Prediction on Spatio-Temporal Features

In [21,22], temporal features were only considered in statistical methods, such as the Kalman filtering, auto-regressive integrated moving average (ARIMA) model and its variants. Machine learning models in [23,24] were carried out to capture the non-linear features of historical traffic data, yet posing challenges in handling the issues of shallow architectures, manual feature selection, and separate learning [25]. Further, deep learning is used for predicting the traffic flow, demonstrating its advantages in comprehending the complex data and capturing critical features accurately [26,27]. Nevertheless, these works rely on the independent traffic flow information, without concerning the correlation between time series. In this light, the long short-term memory network (LSTM), RNN, and gated recurrent unit (GRU) were further proposed to conquer the temporal dependency and solve short-term memory issues [11,28,29]. It is acknowledged that the traffic flow links with historical data and exhibits the spatial interconnectedness, especially in the context of an intricate road network structure [30,31]. In particular, the ST-ResNet was proposed in [32], which represents the road network as a graph, to model the spatial topology. Kipf et al. in [33] used the graph convolutional network (GCN) to further depict topological relationships in the feature extraction and propagation. Additionally, hybrid models have also been proposed, e.g., architectures combining a CNN and LSTM [34], or models integrating deep convolutional neural networks (DCNNs) with LSTM (known as SRCN) [35], which can comprehensively characterize the complexity and diversity of urban traffic.

As mentioned above, the hybrid traffic flow prediction model can enable a more accurate prediction of the contributions of different types of vehicles to the traffic flow, thus providing more precise results. However, recall that current graph-based models focus solely on the geographical locations or physical connections of roads, yet neglect the spatial heterogeneity between different roads and traffic flow patterns.

### 2.2. External Features-Assisted Traffic Flow Prediction

Environmental factors are closely related to the risk of traffic accidents and road congestion, and so it is vital to understand these relationships for predicting traffic flows. Peng et al. in [36] considered the seasonal information with the ARIMA model. Besides, a neural wavelet model was designed in [37] by incorporating the weather data as input to demonstrate the effectiveness of the proposed model. In order to achieve the data fusion of historical traffic flow and weather conditions, the GRU, CNN, and their assembly have been explored for offering potential solutions for the hybrid prediction model [38,39]. It is acknowledged that weather conditions have a uniform impact across the city, which is insufficient for accurately capturing traffic flow variations on specific streets.

Therefore, Blagoiev et al. in [40] evidenced the correlation between traffic flow and air quality, showing that the exhaust emissions generated by traffic flows are recognized as a significant pollution source in the urban air quality. Conversely, changes in the air quality also impact the traffic flow and vehicle operations. Thus, the air quality is related to the traffic flow. Besides, it was demonstrated that the air quality can serve as auxiliary information to obtain more precise predictions [41]. In particular, in 5G communications, the IoV technique (integrating edge computing [42] with digital twin [43]) could monitor traffic flow and the acquisition of air quality data, and the data is transmitted via onboard communication systems to centralized servers or cloud platforms for instantaneous processing, thereby offering possibilities for incorporating the air quality into traffic flow prediction.

In aforementioned works, air pollution conditions are rarely taken as an external factor for traffic flow [44]. In this light, we concentrate on the integration of air quality with spatio-temporal data, and further reveal the correlations between different air pollutants and traffic levels in specific streets.

## 3. Problem Definition and Methodology

### 3.1. Problem Definition

We introduce the method to attain precise traffic volume forecasting within a defined time period while taking into account the historical data on traffic speed, air quality, as well as road networks.

#### 3.1.1. Road Topology

We model the road topology as an indirect graph, $G=\left(V,E\right)$, in which the adjacency between road segments is expressed. $V=\left\{v_{1},v_{2},v_{3},\cdots ,v_{n}\right\}$ denotes the vertex set (representing urban roads), with *n* as the vertices count. Likewise, $E=\left\{e_{1},e_{2},e_{3},\cdots ,e_{m}\right\}$ denotes the edge set, with *m* as the total edge count. An adjacency matrix *A* is further derived on G to represent the adjacency relationship between any pair of roads in the IoV. For the unweighed matrix *A*, 1 indicates that the selected pair of road segments is adjacent, and 0 otherwise.

#### 3.1.2. Traffic Feature Matrix

We regard the traffic velocity as an intrinsic feature of each road segment, denoted as matrix X, where the traffic speed of the *i*-th road at time step *t* is written by $x_{i}^{t}$.

#### 3.1.3. External Feature Matrix

Further, we model external factors as the attribute matrix $K=\left\{K_{1},K_{2},K_{3},\cdots ,K_{l}\right\}$, with *l* as the total number of external factors. We next define a set of type *j*-specific auxiliary information, i.e., $K_{j}=\left\{j^{1},j^{2},j^{3},\cdots ,j^{t}\right\}$. Then, at the *i*-th segment of time step *t*, the *j*-th auxiliary information is denoted as $j_{i}^{t}$.

Next, traffic conditions are inclined to be predicted at the next multi-time steps by setting function *f* with parameters of road topology G, adjacency matrix *A*, traffic feature matrix X, and attribute matrix K, i.e.,(1)$$\left[x^{t-T},x^{t-T+1},\cdots ,x^{t}\right]=f\left(V,E,A,X|K\right).$$

### 3.2. Framework of SHGCN

In Figure 2, the air pollutants, traffic flow information, and spatial data are taken into our framework. More especially, we can decide whether or not to mask the traffic information and relationships between edges (connecting road segments) using a Bernoulli distribution to assess the degree of heterogeneity between road segments. Meanwhile, the *K*-means method is used to select air pollutants highly correlated with the traffic flow. Finally, the prediction is conducted on the assembly of the GCN and GRU.

### 3.3. Spatial Heterogeneity

Details of assessing the spatial heterogeneity are provided in terms of the heterogeneity detection and topology enhancement, respectively.

#### 3.3.1. Heterogeneity Detection

To retain the contextual information within the traffic flow graph, it is designed to capture the temporal patterns in traffic data as well as the correlations between them across various time steps and segments. For encoding these temporal traffic patterns, we utilize temporal convolutions followed by a gating mechanism (i.e., a gated linear unit, GLU) [45], by integrating these patterns into each road segment as follows:(2)$$\left(B^{t-T_{\mathrm{out}}},\cdots ,B^{t}\right)=\mathrm{TC}\left(x^{t-T},\cdots ,x^{t}\right),$$
where $B^{t}\in R^{N\times D}$ represents the road segment embedding matrix at time step *t*, in which the *n*-th column $b_{n}^{t}\in R^{D}$ corresponds to the embedding of road segment $v_{n}$. The embedding dimension is denoted as *D*, and the length of the embedded sequence output after the convolution is denoted as $T_{\mathrm{out}}$ in the temporal convolution (TC) encoder. The temporal convolution kernel size in the TC encoder is three.

In particular, a two-aspect topology enhancement scheme is designed on the graph f(V,E,A,X), consisting of traffic flow enhancement and graph enhancement. First, the traffic flow enhancement is to eliminate the connections between weakly correlated traffic flows across different road segments. Second, the graph enhancement is to remove edges between weakly correlated yet adjacent road segments or to add edges between strongly correlated but non-adjacent segments. Both stages adjust the edges of the graph by involving traffic patterns between road segments.

More especially, consider the heterogeneity detection for road segments. For a segment $v_{n}$, its embedding sequence $\left(b_{n}^{t-T},\cdots ,b_{n}^{t}\right))$ within *T* time steps is obtained from the overall embedding of $\left(B^{t-T},\ldots ,B^{t}\right)$, i.e.,(3)$$p_{n}^{\tau }={\left(b_{n}^{\tau }\right)}^{⊤}\cdot \omega_{0},$$
and(4)$$\;u_{n}=\sum_{\tau =t-T}^{t}p_{n}^{\tau }\cdot b_{n}^{\tau },$$
where $u_{n}$ represents the aggregated representation based on the derived aggregation weights $p_{n}^{\tau }$ over $v_{n}$ at different time steps. Herein, a time step ranging from t−T to *t* is taken and recorded as τ. The aggregation weight $b_{n}^{\tau }$ reflects traffic pattern correlations, with its transpose ${\left(b_{n}^{\tau }\right)}^{⊤}$ enabling dimension-specific weighting through dot product with the learnable parameter vector $\omega_{0}\in R^{D}$. This vector automatically adapts via gradient descent to capture varying feature dimension contributions to temporal weights. The term $p_{n}^{\tau }={\left(b_{n}^{\tau }\right)}^{⊤}\cdot \omega_{0}$ computes temporal attention scores that dynamically adjust to traffic patterns (e.g., peak/non-peak differences). The resulting $u_{n}$ encodes $v_{n}$’s dynamic state over [t−T,t] as an attention-weighted sum of historical embeddings, where high-weight moments correspond to critical transitions like congestion onset or dissipation.

Finally, utilize the cosine similarity coefficient to estimate the degree of heterogeneity between two road segments to reflect the difference in their traffic distribution over time, i.e.,(5)$$q_{m,n}=\frac{u_{m}^{⊤}u_{n}}{∥u_{m}∥∥u_{n}∥},$$
which can measure the correlation level between traffic flow patterns of segments $v_{m}$ and $v_{n}$.

#### 3.3.2. Topology Enhancement

The cosine similarity degree is adopted to estimate heterogeneity between two road segments, reflecting the difference in traffic flow distribution across different periods, and thus distinguishing transportation modes between them. As illustrated in Figure 3, the real-world road network topology and traffic flow information are first input into a traffic feature matrix, where an edge exists between segments A and B once they are adjacent, while there exists no edges between road segments A and D. Then, the similarity of traffic patterns is evaluated via calculating the masking probabilities. In particular, although road segments A and B are adjacent, they have a low correlation in the traffic flow, thus removing the edge between them. Conversely, road segments A and D are connected in the enhanced topology due to the similar traffic patterns with high masking probability. Finally, repeating it for each road segment, we rebuild an enhanced feature matrix that exploits the correlations of traffic patterns instead of the road adjacency.

##### Traffic Flow Enhancement

In the traffic tensor, the enhancement operator $X_{t-T,t}$ (encompassing all historical traffic flows) is devised to characterize the pattern connectivity of each road segment, and to serve as a reference for the traffic flow shielding. As such, the masking probability can be derived via the Bernoulli distribution, i.e., $\rho_{n}^{\tau }∼\mathrm{Bern}\left(1-p_{n}^{\tau }\right)$. Afterwards, we further mask the lower correlated traffic volumes at time step τ of road segment $v_{n}$ to enhance the model generalizability, and the enhanced data by the traffic-level augmentation is denoted as ${\hat{X}}_{t-T,t}$.

##### Graph Enhancement

The enhancement of the traffic flow stems from temporal features, while that at the graph topology pertains to spatial features. Topology-level enhancement is performed on the regional traffic flow graph G to eliminate connections between regions with low interrelated traffic patterns, thereby improving the model’s generalization capability across different regions. Likewise, the enhancement also utilizes a Bernoulli distribution as the basis for shielding. For two adjacent road segments $v_{m}$ and $v_{n}$, if their traffic patterns are not highly correlated, then their connecting edge $\left(v_{m},v_{n}\right)\in E$ will be masked, with the heterogeneity degree $q_{m,n}$ to quantify the traffic pattern difference between two regions. More especially, the masking probability $\rho_{m,n}$ obeys the Bernoulli distribution, i.e., $\rho_{m,n}∼\mathrm{Bern}\left(1-q_{m,n}\right)$, provided that traffic patterns between road segments $v_{m}$ and $v_{n}$ are positively correlated; otherwise, they are directly masked. For any two non-adjacent road segments, the lower degree of heterogeneity $q_{m,n}$ would add an edge between $v_{m}$ and $v_{n}$, based on the masking probability $p_{n}^{\tau }$. Thus, the topology enhancement operation might substantially alter the topology structure.

Next, after two steps in the enhancement, the overall topology enhancement is attained as(6)$$\hat{f}\left(V,\hat{E},\hat{A},\hat{X}|K\right),$$
where ${\hat{X}}^{t-T,t}$ represents the enhancement in traffic flow, and both $\hat{E}$ and $\hat{A}$ represent the enhancements on the graph, respectively.

Since the adjacency matrix $\hat{A}$ encompasses both the vertex set V and edge set $\hat{E}$, (Equation 6) can be further simplified as(7)$$\hat{f}\left(\hat{A},\hat{X}|K\right).$$

### 3.4. External Factors

#### 3.4.1. Incorporating External Features

An augmented matrix method [46], which integrates the spatio-temporal GCN with attribute-enhanced units, is used to propose a traffic prediction model involving the external information (e.g., air quality). Besides, the *K*-means clustering is utilized to identify the air pollutants most correlated with the traffic flow. Note that the injected external feature is dynamic, leading to the traffic feature matrix *X* and attribute matrix *K*.

The dynamic attributes are represented as(8)$$K^{t-T,t}=\left[K^{t-T},K^{t-T+1},K^{t-T+2},\cdots ,K^{t}\right].$$
Therefore, the enhanced matrix involving the external features and traffic feature information at time *t* becomes as(9)$$P_{t}=\left[{\hat{X}}_{t},K^{t-T,t}\right].$$

Lastly, using the enhanced matrix *P* as the input of model *f* would yield the prediction as $\hat{y}$, i.e.,(10)$$\hat{y}=f\left(\hat{A},P\right).$$

Especially, the traffic flow dataset and air quality dataset we adopted exhibit spatio-temporal alignment, thereby fully supporting the integration of air quality features through the aforementioned method.

#### 3.4.2. Filtering Air Components Correlated with Traffic Flow

The factors influencing road air quality include not only vehicular but also meteorological and industrial emissions. To thoroughly understand the environmental conditions of target roadway segments, it is imperative to identify air components which are highly correlated. Such a holistic analysis facilitates the accurate assessment of the environmental status.

Recall that in the above steps, we employ clustering, so the *K*-means method is used to conduct the clustering analysis on air pollutants and compare the results with those of traffic flow clustering, which helps to determine which type of air pollutants are more significantly influenced by the traffic flow. For the historical traffic dataset $X=\left\{x_{1},x_{2},x_{3},\cdots ,x_{n}\right\}$ and a certain air pollution dataset $Y=\left\{y_{1},y_{2},y_{3},\cdots ,y_{n}\right\}$, a matching matrix $M=\left[m_{ij}\right]\left(1\leq i\leq k,1\leq j\leq q\right)$ is defined, where $m_{ij}$ represents the number of sample points that have the same clustering label in the *i*-th class after clustering X, and the *j*-th class after clustering Y. Finally, a clustering similarity metric [47] $B_{k}$ is defined as(11)$$B_{k}=\frac{T_{k}}{\sqrt{P_{k}Q_{k}}},$$
where $T_{k}$, $P_{k}$, and $Q_{k}$ are respectively defined as(12)$$T_{k}=\sum_{i=1}^{k}\sum_{j=1}^{k}m_{ij}^{2}-n,$$(13)$$P_{k}=\sum_{i=1}^{k}m_{i\cdot }^{2}-n,$$
and(14)$$Q_{k}=\sum_{i=1}^{k}m_{\cdot j}^{2}-n,$$
respectively. It can be observed from (Equation 11) that the value of $B_{k}$ is unrelated to the clustering labels. Besides, from (Equation 12) and (Equation 13), it can be deduced that $T_{k}\leq P_{k}$, so $0\leq B_{k}\leq 1$ holds. $B_{k}=1$ and $B_{k}=0$ indicate the completely similar and the completely dissimilar between the results of two clustering iterations, respectively. In particular, $m_{i\cdot }$, $m_{\cdot j}$, and $m_{\cdot \cdot }$ are, respectively, defined as(15)$$m_{i\cdot }=\sum_{j=1}^{k}m_{ij},$$(16)$$m_{\cdot j}=\sum_{i=1}^{k}m_{ij},$$
and(17)$$m_{\cdot \cdot }=n=\sum_{i=1}^{k}\sum_{j=1}^{k}m_{ij}.$$
Equation (Equation 15) represents the calculation of the number of matching time steps where a specific category of traffic flow clusters aligns with each category of pollutant clusters, whereas (Equation 16) represents the calculation of the number of matching time steps where a specific category of pollutant clusters aligns with each category of traffic flow clusters. By comparing the similar results of traffic flow and air pollution clustering, the relevance between traffic flow and air pollution can be analyzed. The clustering similarity index $B_{K}$’s calculation reveals that it defines a matrix to represent the number of shared instances between a class from the first clustering and each class from the second clustering (similar to finding the intersection of both clustering results), indicating that a class from the first clustering is associated with each class from the second clustering. The clustering similarity index $B_{k}$ cannot directly represent causal relationships between traffic flow and air pollutants but serves as a reference metric for indirectly inferring potential associations between specific air pollutants and traffic patterns.

### 3.5. Spatio-Temporal GCN

The GCN and GRU are used to create a hybrid model *f* [46], thereby capturing spatio-temporal dependencies in the traffic flow. Especially, GCN units are utilized to acquire the representation for roads with the influence of interconnected ones, and $y_{l+1}=\sigma \left({\tilde{D}}^{-\frac{1}{2}}\tilde{A}{\tilde{D}}^{-\frac{1}{2}}y_{l}W_{l}\right)$ represents the modeling, where $\tilde{A}=\hat{A}+I$ indicates the self-connected adjacency matrix, the degree matrix is represented as $\tilde{D}$, and σ denotes the activation function. The GCN is implemented with a single layer, where the filter dimension matches the number of hidden units in the GRU (64 units)—a configuration that was experimentally validated in our studies. Further, $W_{l}$ denotes the weight matrix for the *l*-th convolutional layer, $y_{l}$ represents the output prediction, $y_{0}=P^{t}$ signifies the representation during the spatial–temporal feature modeling at time *t*, and both the enhancement matrix P and road adjacency matrix *A* are used as the inputs of the GCN.

Afterwards, the historical traffic state $h_{t-1}$ is bonded with the current street representation at time step *t*, and then the hidden state $c_{t}$ can be derived. Meanwhile, we determine to discard and identify which new information from $c_{t}$ should be included when deriving the final traffic state $h_{t}$, i.e.,(18)$$u_{t}=\sigma \left(W_{u}\cdot \left[gc\left(P_{t},\hat{A}\right),h_{t-1}\right]+b_{u}\right),$$(19)$$r_{t}=\sigma \left(W_{r}\cdot \left[gc\left(P_{t},\hat{A}\right),h_{t-1}\right]+b_{r}\right),$$(20)$$c_{t}=\tanh\left(W_{c}\cdot \left[gc\left(P_{t},\hat{A}\right),\left(r_{t},h_{t-1}\right)\right]+b_{c}\right),$$
and(21)$$h_{t}=u_{t}\cdot h_{t-1}+\left(1-u_{t}\right)\cdot c_{t},$$
where gc(·) represents the graph convolution operation with learnable parameters *W* and *b*. Finally, the loss function is defined as(22)$$\mathrm{Loss}=∥y_{t}-{\hat{y}}_{t}∥+\lambda \cdot L_{\mathrm{reg}},$$
where $y_{t}$ and ${\hat{y}}_{t}$ separately denote the actual and predicted values, $L_{reg}$ denotes the $L_{2}$ regularization, and λ is a hyperparameter, respectively.

## 4. Experiments and Results

### 4.1. Datasets

Two real-world datasets are used for experiments, i.e.,

Traffic flow data in Aarhus, Denmark—the traffic flow data records the average traffic speed every 5 min. For missing values, we employed interpolation methods to ensure data completeness. Data from 100 road segments were selected from 2 August 2014 to 17 August 2014. The connections between urban roads are modeled by a 100×100 adjacency matrix, which indexes rows by road segments and columns by timestamps. The air pollution data in Aarhus, Denmark—recorded at 5 min intervals synchronized with traffic measurements—consists of pre-computed Air Quality Index (AQI) values generated by the original dataset authors, integrating PM2.5 and PM10 concentrations with other pollutants (ozone, carbon monoxide, sulfur oxides, and nitrogen oxides). By adopting *K*-means clustering of these standardized AQI levels, we identify pollution-traffic correlations. Since the dataset contains no AQI scores exceeding 300, we classify air quality into five AQI classification levels: 0–50 (Good), 51–100 (Moderate), 101–150 (Unhealthy for Sensitive Groups), 151–200 (Unhealthy), and 201–300 (Very Unhealthy). Then, the categorized data is structured into a dynamic 100×4320 attribute matrix for spatio-temporal analysis.Traffic flow data in Newcastle upon Tyne, UK—the traffic flow data records vehicle counts at 5 min intervals. For missing values, we employed interpolation methods to ensure data completeness. Data from 136 road segments were selected from 20–30 June 2025. The connections between urban roads are modeled by a 136×136 adjacency matrix, which indexes rows by road segments and columns by timestamps. The air pollution data in Newcastle upon Tyne, UK—recorded at 5 min intervals and synchronized with traffic flow measurement timestamps—consists of air pollutant concentrations from which we calculated the Air Quality Index (AQI). Our analysis incorporates concentrations of carbon monoxide, ozone, PM2.5, and nitrogen dioxide. For road segments without direct air quality monitoring, we adopted data from the nearest available monitoring locations. The preprocessing methodology for air quality data aligns with that applied to the Aarhus dataset. The categorized data is structured into a dynamic 136×2880 attribute matrix for spatio-temporal analysis.

### 4.2. Evaluation Metrics

The following metrics are employed to evaluate the prediction accuracy.

Root mean square error (RMSE), i.e.,(23)$$\mathrm{RMSE}=\sqrt{\frac{1}{n}\sum_{t=1}^{n}{\left(y_{t}-{\hat{y}}_{t}\right)}^{2}},$$
which can represent the model performance, and a smaller value would indicate a better model with the higher accurate prediction level.Mean absolute error (MAE), i.e.,(24)$$\mathrm{MAE}=\frac{1}{n}\sum_{t=1}^{n}|y_{t}-{\hat{y}}_{t}|,$$
where the smaller the MAE value is, the better the performance of the prediction model becomes.Accuracy, i.e.,(25)$$\mathrm{Accuracy}=1-\frac{∥y-\hat{y}{∥}_{F}}{{∥y∥}_{F}},$$
in which as the value of the Frobenius norm ${∥\cdot ∥}_{F}$ gets closer to one, the prediction becomes more accurate.

### 4.3. Parameter Settings

Our proposed model, the SHGCN, is optimized using Adam for training, with a learning rate of 0.001, a batch size of 64, and a training set proportion of 0.8. Using the Aarhus dataset as an illustrative example, we initially evaluate training epochs from the set [100, 500, 1000, 1500, 2000, 2500] to observe their impact on model performance. Figure 4 shows that, as the training epochs increase, the evaluation metrics would exhibit stability, with a significant inflection point observed at 1000 epochs and the stabilization at 1500 epochs. Subsequently, with the training epochs fixed at 1500, the number of hidden units is selected from [16, 32, 64, 100, 128]. Figure 5 demonstrates that the stability in evaluation metrics is achieved when the number of units is 64. Therefore, to mitigate the potential overfitting, the training epochs are selected as 1500, and the number of hidden units is established as 64. Moreover, the data is divided into training and test sets with an approximate ratio of 8:2.

### 4.4. Baselines

The proposed SHGCN model is compared with the following baselines: (1) Historical Average (HA) [48], (2) Support Vector Regression (SVR) [49], (3) Graph Convolutional Network (GCN) [33], (4) Gated Recurrent Unit (GRU) [29], (5) Diffusion Convolutional Recurrent Neural Network (DCRNN) [50], (6) Temporal Graph Convolutional Network (TGCN) [16], and (7) Multi-Scale Dynamic Residual Network (MSDR) [51].

### 4.5. Experiment Results

The experiment is designed to explore three aspects: the impact of the introduced information on different air pollution components, the prediction accuracy comparison with the baseline, and ablation experiments.

#### 4.5.1. Clustering Results of Air Pollutants and Traffic Flow

To filter out the air pollutants most correlated with the traffic flow, two clustering analyses are conducted on both the traffic flow and air pollutants. By comparing the similarity of the clustering results (where a higher value of $B_{k}$ signifies a stronger association between the pollutant component and traffic flow), we find that carbon monoxide exhibits the highest correlation with traffic flow during the learning period in Table 1.

In Table 2, we conduct a set of control experiments and establish another control group without integrating air components. Among them, the group incorporating CO performs the best, consistent with the clustering results. Besides, some pollutant components, such as $O_{3}$, exhibit a relatively low correlation with the traffic flow. Further, the integration of $O_{3}$ has a lower prediction accuracy than the control group without air quality integration, indicating that a correlation $B_{k}$ higher than a certain threshold is necessary to improve the prediction results.

Subsequently, in the Newcastle dataset (as presented in Table 3 and Table 4), the experimental group incorporating CO data demonstrates an optimal performance—a finding consistent with both the clustering analysis results and the performance patterns observed in other experimental groups. Our experimental results demonstrate that incorporating CO, $O_{3}$, or PM2.5 concentrations consistently improves the prediction performance across 15 min, 30 min, and 60 min prediction windows, as measured by the RMSE, MAE, and accuracy metrics. In contrast, models integrating ${\mathrm{NO}}_{2}$ showed a degraded performance compared to the baseline without air quality inputs. These findings reaffirm that only air pollutants exhibiting strong correlations with traffic flow patterns can enhance prediction accuracy. Notably, this CO-integration superiority is replicated in both the Aarhus and Newcastle datasets, suggesting that vehicular emissions may constitute the predominant source of urban CO pollution.

#### 4.5.2. Comparison Baselines

To validate the effectiveness of the SHGCN, its comparison with baseline models over various prediction horizons is illustrated in Table 5 and Table 6.

As exemplified in Table 5, the experiment evaluates the performance of the SHGCN across different prediction time intervals, including 15 min, 30 min, and 60 min, respectively. The results demonstrate that the deep-learning-based methods (i.e., the SHGCN, GCN, GRU, DCRNN, TGCN, and MSDR) outperform other ones in terms of the prediction accuracy. Within a 15 min prediction window, and compared to the HA and SVR, the SHGCN can reduce the RMSE by approximately 38.19% and 15.76%, respectively, and the MAE by approximately 25.03% and 12.84%, respectively. From a spatio-temporal perspective, the SHGCN can reduce the RMSE by approximately 11.95% and 9.76% compared to the GCN and GRU separately. Compared to the method that only considers the single relationship in space or time, the MAE can also be lowered by approximately 11.30% and 10.08%, respectively. More importantly, by involving air quality factors, the SHGCN also outperforms existing hybrid models, i.e., the DCRNN, TGCN, and MSDR, with an RMSE reduction of approximately 8.08%, 4.58%, and 4.31% and a MAE reduction of approximately 9.07%, 4.05%, and 2.91%, respectively.

Within a 30 min prediction window, the SHGCN could reduce the RMSE by approximately 20.28% and 13.30%, and the MAE by approximately 19.73% and 10.12%, as compared to the HA and SVR, respectively. From a spatio-temporal perspective, the SHGCN could reduce the RMSE by approximately 11.12% and 8.67%, and the MAE by approximately 9.94% and 8.22%, compared to the GCN and GRU, respectively. Considering air quality factors, the SHGCN outperforms the DCRNN, TGCN, and MSDR with an RMSE reduction of approximately 7.78%, 4.55%, and 4.09% and an MAE reduction of approximately 7.16%, 6.62%, and 4.92%, respectively.

Within a 60 min prediction window, the SHGCN reduces the RMSE by approximately 11.63% and 12.46% and the MAE by approximately 15.65% and 10.86% compared to the HA and SVR, respectively. From a spatio-temporal perspective, the SHGCN reduces the RMSE by approximately 10.43% and 7.67% and the MAE by approximately 10.71% and 9.22% compared to the GCN and GRU, respectively. Considering air quality factors, the SHGCN outperforms the DCRNN, TGCN, and MSDR, with an RMSE reduction of approximately 7.35%, 5.05%, and 4.67% and an MAE reduction of approximately 9.03%, 5.73%, and 5.15%, respectively. It is shown that the SHGCN exhibits superiority, with a better forecasting performance and long-term predictive capabilities across different horizons.

As shown in Table 6, the SHGCN achieves an 8.63–41.26% improvement over existing models in both the RMSE and MAE metrics on the Newcastle dataset, further demonstrating its superior predictive performance. Notably, the SHGCN demonstrates significant improvements over traditional single-prediction models and outperforms existing advanced deep learning models, achieving RMSE reductions of 17.97%, 15.81%, and 9.69% compared to the DCRNN, TGCN, and MSDR at the 15 min prediction window; 16.26%, 8.85%, and 9.10% at the 30 min window; and 20.44%, 13.87%, and 8.63% at the 60 min window, respectively, further confirming its superior performance in prediction tasks.

#### 4.5.3. Ablation Experiments

Ablation experiments demonstrate that both air quality factors and topology enhancement could impact the traffic prediction. The experimental settings are divided into three scenarios, i.e, one where only air quality factors are integrated, one where only the topology enhancement is applied, and one where both air quality factors and topology enhancement are incorporated. In Table 7, the results with integrated air quality factors are listed in the fourth column, those with the topology enhancement are in the fifth column, and those involving both air quality factors and topology enhancement are in the sixth column.

Within a 15 min forecasting horizon and only involving air quality factors (e.g., CO), the RMSE of the SHGCN is 4.48% and 0.15% lower than that of the DCRNN and TGCN, respectively. Compared to the DCRNN, TGCN, and MSDR, the SHGCN (only enhanced) could reduce the RMSE by 5.64%, 2.05%, and 1.77%, respectively. Although the MSDR achieves a 0.13% reduction in the RMSE compared to the SHGCN (CO), when simultaneously incorporating air quality factors and applying the graph enhancement, the SHGCN (CO + enhanced) model reduces the RMSE by 8.08%, 4.58%, and 4.31%, respectively, compared to the DCRNN, TGCN, and MSDR.

Within a 30 min forecasting horizon and only involving air quality factors, the RMSE of the SHGCN (CO) model is, respectively, 3.39% and 0.01% lower than that of the DCRNN and TGCN. With only the topology enhancement, compared to the DCRNN, TGCN, and MSDR, the SHGCN (enhanced) reduces the RMSE by 4.92%, 1.59%, and 1.12%, respectively. Although the MSDR achieves a 0.47% reduction in the RMSE compared to the SHGCN (CO), when simultaneously incorporating air quality factors and applying the graph enhancement, the SHGCN (CO + enhanced) model exhibits an RMSE reduction of 7.78%, 4.55%, and 4.09%, respectively, compared to the DCRNN, TGCN, and MSDR.

Within a 60 min forecasting horizon and only involving air quality factors, the RMSE of the SHGCN (CO) is, respectively, 2.30% lower than that of the DCRNN. With only the topology enhancement, compared to the DCRNN, TGCN, and MSDR, the SHGCN (enhanced) reduces the RMSE by 3.59%, 1.20%, and 0.80%, respectively. Although the TGCN and MSDR achieve a 0.12% and 0.52% reduction in the RMSE compared to the SHGCN (CO), when simultaneously incorporating air quality factors and applying the graph enhancement, the SHGCN (CO + enhanced) model exhibits the RMSE reduction of 7.35%, 5.05%, and 4.67%, respectively, compared to the DCRNN, TGCN, and MSDR.

It is worth noting that when simultaneously incorporating both air quality factors and the graph enhancement, the model dominates that with only the single heterogeneity information. In particular, the prediction errors are reduced by 4.43% and 2.58% within a 15 min prediction horizon, by 4.55% and 3.01% within a 30 min horizon, and by 5.17% and 3.90% within a 60 min horizon, indicating that the temporal and spatial heterogeneity are mutually complemented. In particular, while the baseline TGCN model employs a GRU for temporal feature extraction and a GCN for spatial feature extraction, the SHGCN similarly utilizes a GRU-GCN architecture. This parallel structure demonstrates that either air quality factors alone or graph enhancement alone can improve the prediction performance, but their synergistic integration yields optimal results.

Similarly, the results in Table 8 demonstrate that both air quality factors and topological enhancement influence traffic flow prediction. In this dataset, both the individual incorporation of air quality factors and the standalone topological enhancement demonstrate a superior predictive performance compared to the GCN+GRU-based TGCN model, reaffirming that either approach—integrating air quality parameters or implementing topological enhancement alone—can effectively improve forecasting accuracy. And, the SHGCN (CO+enhance) model demonstrates a superior predictive performance compared to both the SHGCN (CO) and SHGCN (enhance), achieving RMSE reductions of 11.17% and 10.59%, respectively, at the 15 min prediction horizon, 5.93% and 1.32% at the 30 min horizon, and 8.59% and 3.12% at the 60 min horizon. Specifically, either air quality factors alone or topological enhancement alone can improve the prediction performance, while their synergistic integration yields optimal results.

### 4.6. Interpretation of SHGCN

The explanation of the SHGCN is divided into two parts: the prediction horizon, as well as the impact of topology enhancement and air quality factors on prediction accuracy.

#### 4.6.1. Long-Term and Short-Term Forecasting

We select a road from the dataset in Aarhus, Denmark. We give the visualization of 6 h predictions, 24 h predictions, and predictions for the entire dataset in Figure 6, Figure 7 and Figure 8, respectively. The horizontal and vertical axes represent the prediction duration and traffic speed, respectively. Further, the red lines denote predicted values, while the blue lines denote actual values.

As depicted in Figure 6, the trend prediction is relatively accurate within the first 15 min, with predicted values closely aligned with the actual values. However, after the prediction duration reaches 1 h, some discrepancies emerge between the predicted and actual values. Nevertheless, the SHGCN could effectively capture the changing trends in traffic flow. The short-term forecasting performance may be attributed to the topology enhancement and adjustments made to the relationships between road segments, coupled with the relatively limited influencing factors within a short time-frame.

Figure 7 illustrates that the discrepancy between predicted and actual values arises over a longer prediction period. Yet, the overall trend performs similarly, and predictions are relatively accurate during some time intervals. That might be attributed to the periodic nature of traffic flow, to which the SHGCN adapts, thereby enhancing the model’s performance during some specific time intervals.

In Figure 8, noticeable deviations exist between predicted and actual values for a very long prediction duration, possibly due to the complexity of long-term prediction, where the model may struggle to adapt to sudden changes. However, due to the inclusion of air quality factors, where external elements act as the supplementary information to improve the performance effectively.

As described above, the SHGCN struggles to capture the inflection points in the speed variation trend, resulting in significant deviations between the predicted and actual values. The discrepancy may arise from the abrupt change, impacted by the spatial heterogeneity and air quality factors, as well as other contributing factors.

#### 4.6.2. Importance of Incorporating Topology Enhancement and Air Quality Factors

We also illustrate the ablation results to evaluate the impact of dynamic external data. Figure 9 exhibits the visualization results by considering various factors, showing that the integration of topology enhancement with air quality factors can perform better. Moreover, Figure 10 shows the visualization results by incorporating different air pollution components, where each component has varying degrees of correlation with the traffic flow, contributing to the prediction. In particular, in some specific periods, the prediction may not align with the clustering results due to the significant variations in clustering results in the proximity. The results indicate that air quality factors are more sensitive to changes in urban traffic flow, providing a more precise trend prediction on individual streets.

## 5. Conclusions

In this work, the urban traffic flow prediction problem was investigated in VANETs, whereby the spatial heterogeneity incurred by traffic patterns and the temporal heterogeneity brought by environmental factors were jointly considered. In particular, we first proposed the SHGCN prediction model to optimize the road topology, and then detected the heterogeneity of traffic information stored in the TC encoder. More especially, the air pollutants highly correlated with the traffic flow were filtered out via the *K*-means clustering method, and then were incorporated into the traffic flow information as an enhanced matrix. Experimental results demonstrated that the SHGCN achieves a 2.91–41.26% lower RMSE and MAE compared to baseline models. Ablation experimental results verified that incorporating both air quality factors and traffic patterns improves the prediction performance to varying degrees, confirming its effectiveness in capturing the correlations among air pollutants, traffic flow, and road topology. Our method effectively integrates dynamic external factors but shows limitations in processing stochastic events (e.g., accidents), which lack consistent spatio-temporal correlations with traffic patterns. This limitation represents a significant research challenge that necessitates further model refinement and constitutes a primary focus for our future work.

## Figures and Tables

**Figure 1 sensors-25-05127-f001:**
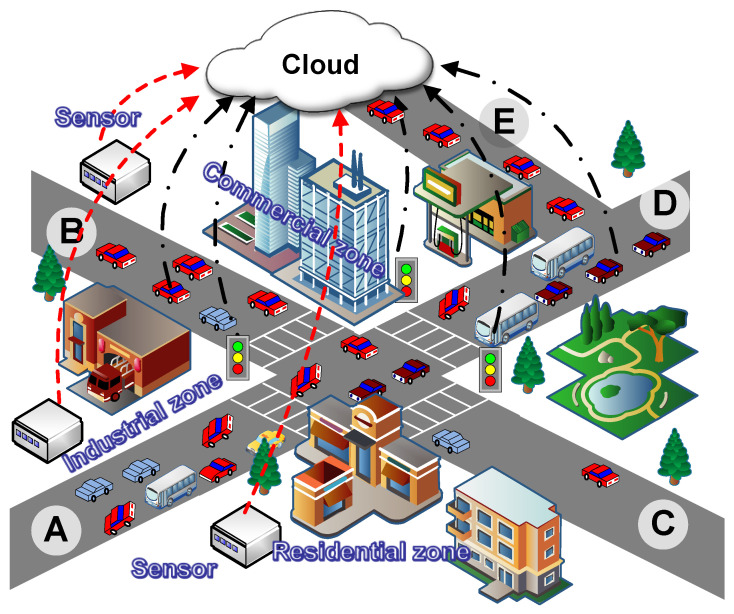
An example that illustrates the relationship between the traffic flow, road topology, and environmental factors. Different regions exhibit distinct traffic patterns, which are further influenced by localized air pollution components with strong regional characteristics.

**Figure 2 sensors-25-05127-f002:**
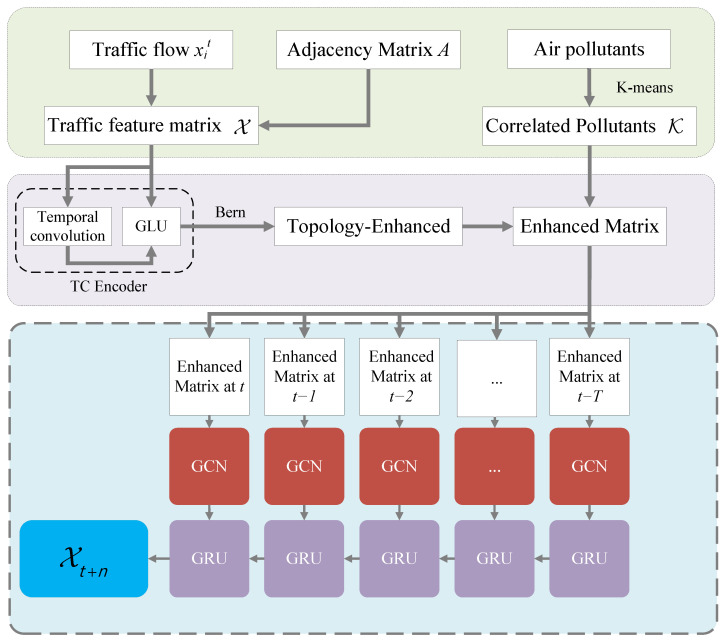
Overall frame diagram of the SHGCN model. The framework includes data preprocessing, graph enhancement, traffic information enhancement, and prediction parts, respectively.

**Figure 3 sensors-25-05127-f003:**
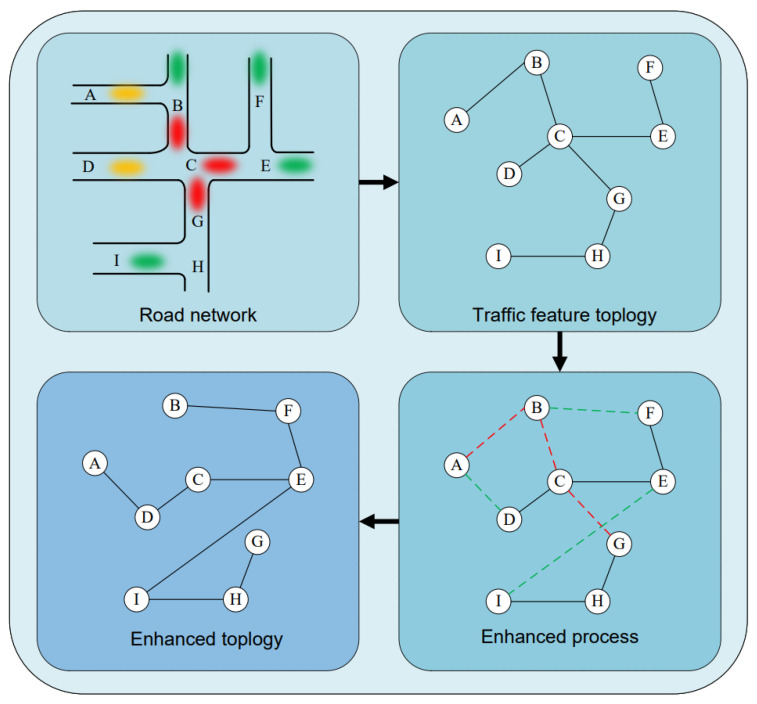
Illustration of the enhancement of road topology. The vertices represent streets, and the edges indicate traffic correlations between vertices. This method filters out connections between less correlated road segments while adding connections between highly correlated ones.

**Figure 4 sensors-25-05127-f004:**
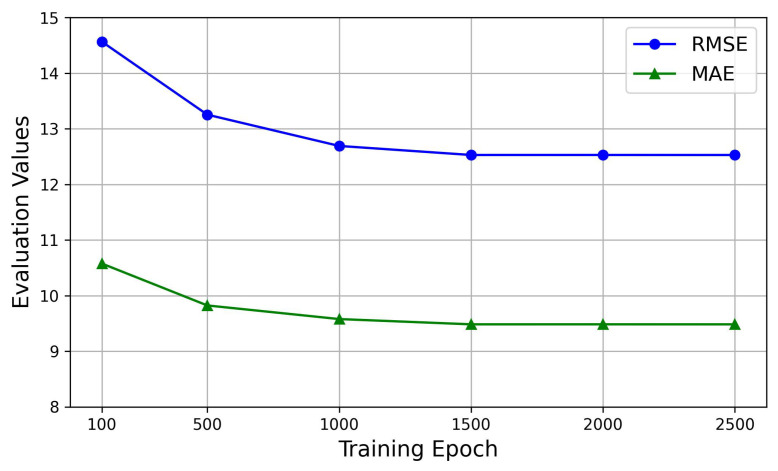
The impact of epoch selection on prediction performance.

**Figure 5 sensors-25-05127-f005:**
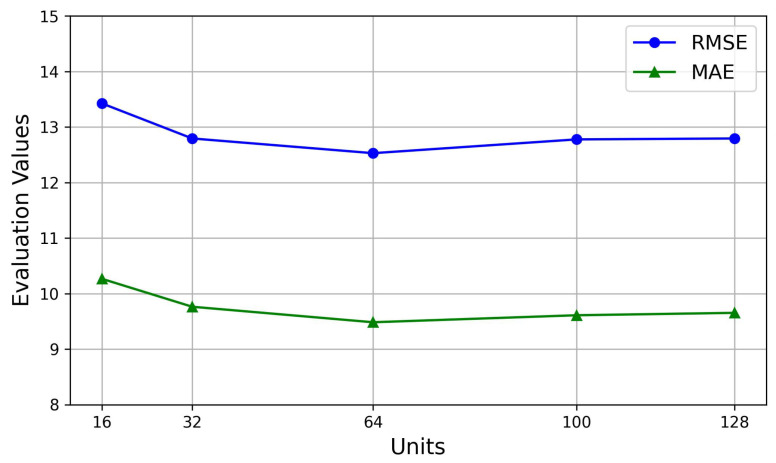
The impact of units selection on prediction performance.

**Figure 6 sensors-25-05127-f006:**
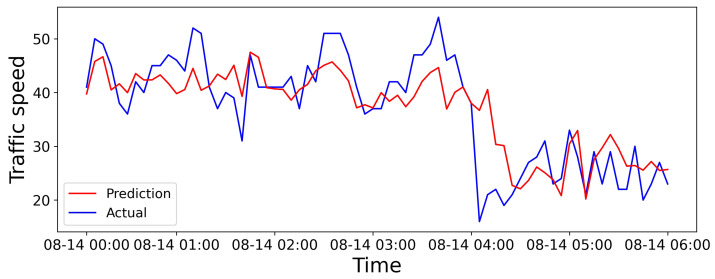
Visual results over the 6 h prediction horizon.

**Figure 7 sensors-25-05127-f007:**
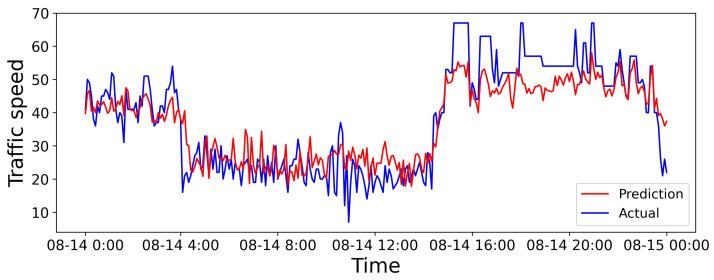
Visual results for the 24 h prediction horizon.

**Figure 8 sensors-25-05127-f008:**
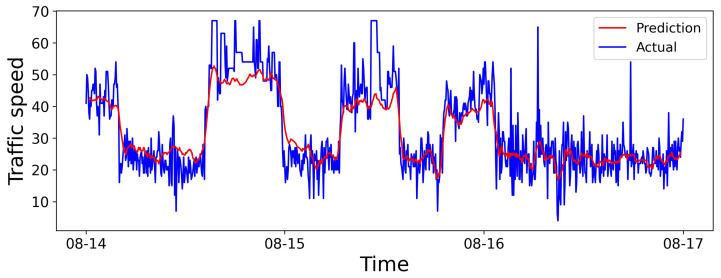
Visual results for the prediction horizon.

**Figure 9 sensors-25-05127-f009:**
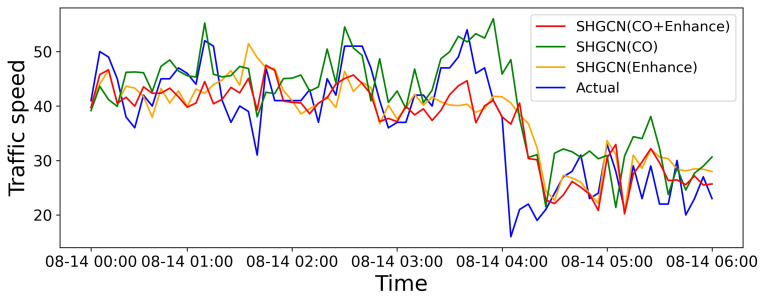
Comparison between results with and without considering heterogeneous factors.

**Figure 10 sensors-25-05127-f010:**
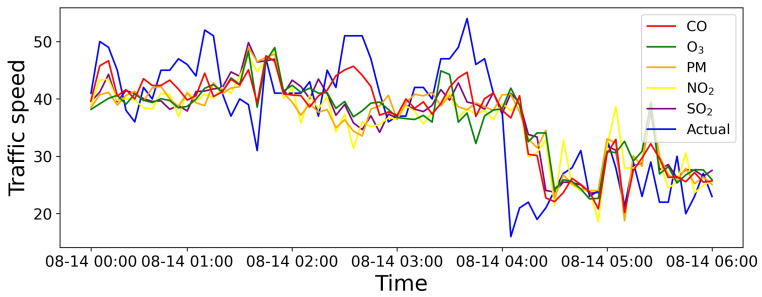
Comparison between results with and without incorporating air pollution components.

**Table 1 sensors-25-05127-t001:** Clustering results of air pollution components in the Aarhus dataset. Carbon monoxide has the highest correlation with traffic flow.

		Aarhus			
Cluster Similarity Metric	CO	$O_{3}$	${\mathrm{SO}}_{2}$	PM	${\mathrm{NO}}_{2}$
$B_{k}$	0.5169	0.2182	0.2603	0.2619	0.3497

**Table 2 sensors-25-05127-t002:** Performance comparison of models incorporating various air pollutants in the Aarhus dataset.

				Aarhus			
Time	Metric	CO	$O_{3}$	${\mathrm{SO}}_{2}$	PM	${\mathrm{NO}}_{2}$	Not Integrated
	RMSE	12.5275	13.1304	13.1021	13.1103	12.7745	12.8598
15 min	MAE	9.4828	9.8854	9.8801	9.8763	9.6492	9.6927
	Accuracy	0.7912	0.7757	0.7768	0.7766	0.7863	0.7842
	RMSE	13.3529	14.0531	14.0364	14.0223	13.7028	13.7672
30 min	MAE	10.1531	10.8237	10.8168	10.8097	10.3726	10.4289
	Accuracy	0.7775	0.7603	0.7615	0.7620	0.7722	0.7701
	RMSE	14.8012	15.7129	15.6859	15.6694	15.4019	15.4021
60 min	MAE	10.6697	11.6064	11.5902	11.5891	11.0765	11.0827
	Accuracy	0.7631	0.7459	0.7466	0.7474	0.7558	0.7554

**Table 3 sensors-25-05127-t003:** Clustering results of air pollution components in the Newcastle dataset. Carbon monoxide has the highest correlation with traffic flow.

		Newcastle		
Cluster Similarity Metric	CO	$O_{3}$	PM2.5	${\mathrm{NO}}_{2}$
$B_{k}$	0.5185	0.4024	0.3333	0.2708

**Table 4 sensors-25-05127-t004:** Performance comparison of models incorporating various air pollutants in the Newcastle dataset.

			Newcastle			
Time	Metric	CO	$O_{3}$	PM2.5	${\mathrm{NO}}_{2}$	Not Integrated
	RMSE	13.0786	13.6331	14.2051	15.2487	14.6272
15 min	MAE	8.4707	8.7934	9.2434	9.8574	9.2638
	Accuracy	0.8331	0.8260	0.8186	0.8055	0.8104
	RMSE	14.2845	14.6017	14.4008	15.4207	15.0536
30 min	MAE	9.2185	9.2416	9.2973	10.0594	10.0118
	Accuracy	0.8178	0.8129	0.8122	0.8035	0.8096
	RMSE	14.7128	14.9965	15.1736	16.1318	15.1864
60 min	MAE	9.4474	9.3441	9.6611	10.3876	9.7246
	Accuracy	0.8126	0.8065	0.8062	0.7941	0.8040

**Table 5 sensors-25-05127-t005:** Comparison of performance between SHGCN and baselines with various prediction horizons in the Aarhus dataset.

					Aarhus				
Time	Metric	HA	SVR	GCN	GRU	DCRNN	TGCN	MSDR	SHGCN
	RMSE	16.7498	14.8709	14.2274	13.8823	13.6291	13.1284	13.0921	12.5275
15 min	MAE	12.6490	10.8795	10.6903	10.5462	10.4286	9.8827	9.7672	9.4828
	Accuracy	0.7406	0.7630	0.7657	0.7703	0.7722	0.7763	0.7791	0.7912
	RMSE	16.7498	15.4021	15.0238	14.6197	14.4801	13.9901	13.9229	13.3529
30 min	MAE	12.6490	11.2958	11.2737	11.0621	10.9359	10.8726	10.6782	10.1531
	Accuracy	0.7406	0.7482	0.7520	0.7562	0.7578	0.7605	0.7642	0.7775
	RMSE	16.7498	16.9085	16.5239	16.0308	15.9762	15.5892	15.5265	14.8012
60 min	MAE	12.6490	11.9698	11.9501	11.7536	11.7289	11.3184	11.2494	10.6697
	Accuracy	0.7406	0.7352	0.7411	0.7422	0.7432	0.7481	0.7501	0.7631

**Table 6 sensors-25-05127-t006:** Comparison of performance between SHGCN and baselines with various prediction horizons in the Newcastle dataset.

					Newcastle				
Time	Metric	HA	SVR	GCN	GRU	DCRNN	TGCN	MSDR	SHGCN
	RMSE	19.8753	17.4312	17.1533	16.6129	15.9431	15.5338	14.4815	13.0786
15 min	MAE	13.1284	12.3486	12.0562	11.7281	10.1156	10.1891	9.8861	8.4707
	Accuracy	0.7531	0.7749	0.7795	0.7813	0.7963	0.8017	0.7985	0.8331
	RMSE	19.8753	18.9864	18.6245	17.9531	17.0582	15.6713	15.7153	14.2845
30 min	MAE	13.1284	12.9515	12.5631	12.3124	10.1219	10.3431	10.2541	9.2185
	Accuracy	0.7531	0.7583	0.7612	0.7685	0.7815	0.8001	0.8111	0.8178
	RMSE	19.8753	21.2533	20.5582	19.9824	18.4936	17.0829	16.1024	14.7128
60 min	MAE	13.1284	16.0821	15.7498	15.1532	13.1481	11.4869	10.8135	9.4474
	Accuracy	0.7531	0.7225	0.7288	0.7345	0.7588	0.7824	0.7952	0.8126

**Table 7 sensors-25-05127-t007:** Ablation experiments under different experiment settings in the Aarhus dataset.

				Aarhus			
Time	Metric	DCRNN	TGCN	MSDR	SHGCN		
					CO	Enhanced	CO + Enhanced
	RMSE	13.6291	13.1284	13.0921	13.1087	12.8598	12.5275
15 min	MAE	10.4286	9.8827	9.7672	9.8126	9.6927	9.4828
	Accuracy	0.7722	0.7763	0.7791	0.7780	0.7842	0.7912
	RMSE	14.4801	13.9901	13.9229	13.9893	13.7672	13.3529
30 min	MAE	10.9359	10.8726	10.6782	10.8056	10.4289	10.1531
	Accuracy	0.7578	0.7605	0.7642	0.7627	0.7701	0.7775
	RMSE	15.9762	15.5892	15.5265	15.6082	15.4021	14.8012
60 min	MAE	11.7289	11.3184	11.2494	11.2957	11.0827	10.6697
	Accuracy	0.7432	0.7481	0.7501	0.7489	0.7554	0.7631

**Table 8 sensors-25-05127-t008:** Ablation experiments under different experiment settings in the Newcastle dataset.

				Newcastle			
Time	Metric	DCRNN	TGCN	MSDR	SHGCN		
					CO	Enhanced	CO + Enhanced
	RMSE	15.9431	15.5338	14.4815	14.7233	14.6272	13.0786
15 min	MAE	10.1156	10.1891	9.8861	9.4361	9.2638	9.4707
	Accuracy	0.7963	0.8017	0.7985	0.8096	0.8104	0.8331
	RMSE	17.0582	15.6713	15.7153	15.1856	15.0536	14.2845
30 min	MAE	10.1219	10.3431	10.2541	10.0755	10.0118	9.2185
	Accuracy	0.7815	0.8001	0.8111	0.8037	0.8096	0.8178
	RMSE	18.4936	17.0829	16.1024	16.0949	15.1864	14.7128
60 min	MAE	13.1481	11.4869	10.8135	10.1507	9.7246	9.4474
	Accuracy	0.7588	0.7824	0.7952	0.7991	0.8040	0.8126

## Data Availability

The Aarhus datasets analyzed in this study are openly available in the CityPulse Smart City Datasets repository: http://iot.ee.surrey.ac.uk:8080/datasets.html (accessed on 10 August 2025). The Newcastle datasets are similarly available through the Urban Observatory repository: https://newcastle.urbanobservatory.ac.uk/ (accessed on 10 August 2025).

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
