# Peer review of "Spatio-Temporal Heterogeneity-Oriented Graph Convolutional Network for Urban Traffic Flow Prediction"

_sensors, 2025, doi:10.3390/s25165127_

Round 1
Reviewer 1 Report
Comments and Suggestions for Authors
This article presents a framework for monitoring traffic speeds on each road, along with air quality data, to understand how these factors interact and predict future traffic speeds. They connect roads with similar patterns, even if they are geographically far away. GCN is applied to capture the spatial relationships between roads, and GRU is used to model how traffic changes over time, further improving the predictions by combining selected air quality data. The proposed model is tested on real city data, and evaluation results show that spatial heterogeneity and external factors significantly improve prediction accuracy.
Limitations and Concerns:
i. From the contribution listed on page 3, the first point indicates that this is focused on ‘urban traffic flow’, whether the abstract or title seems generalized to any situation. As there may be differences in urban, highway, and other areas' traffic, it is recommended to specify the target/scope of this research more clearly, along with proper reasoning.
ii. They show a correlation between traffic patterns and some pollutants. However, it remains unclear whether air quality directly influences traffic speeds or if both are driven by another factor (e.g., peak hours).
iii. It is not clear how they combine the air quality and traffic data. Traffic data covers many individual road segments, while air quality is typically measured at only a few fixed points in the city and can vary significantly over short distances (e.g., near industrial zones). It is important to explain how they match the limited air quality data to all the roads in the model.
iv. While integrating air quality is important, other factors like weather, geography, events, or incidents can also affect traffic. Can the proposed model include these in the future, and do the authors believe it would improve predictions further?
Reviewer 2 Report
Comments and Suggestions for Authors
Journal: Sensors (ISSN 1424-8220)
Manuscript ID: sensors-3756163
Type: Article
Title: Spatio-Temporal Heterogeneity-Oriented Graph Convolutional Network for Traffic Flow Prediction
Comments to the author(s)
The dataset used in this study is based on data collected between August 2 and August 17, 2014, from 100 road segments. Given that the manuscript is being submitted in 2025, it is quite challenging to align such dated traffic data with current transportation dynamics, infrastructure changes, and technological advancements. While the proposed approach and model are indeed interesting and innovative, the use of decade-old data significantly limits the study's practical relevance and applicability. Therefore, publishing the manuscript in its current form, without incorporating more recent datasets or at least validating the model with updated data, may not be appropriate.
Abstract Evaluation: While the abstract provides a reasonable overview of the methodology, it remains somewhat general in terms of the results. To improve clarity and completeness, it is recommended to include more specific and quantitative findings that reflect the model’s performance and support the stated contributions. This would help readers better understand the practical significance of the proposed approach.
Could the authors please clarify whether Figure 1 was created by the authors themselves or adapted from another source? If it was taken or modified from an existing publication, please provide the appropriate citation. If it is original, a brief statement indicating that the figure was developed by the authors would be helpful for clarity.
The section discussing air pollutants would benefit from a clearer and more specific explanation. While the impact of environmental factors on traffic conditions is an undeniable reality, it would enhance the readers' understanding and interest if the manuscript briefly elaborated on how air pollutants—such as particulate matter, oxides, and sulfides—specifically influence traffic flow patterns or prediction accuracy. Providing a concise explanation of the mechanisms or correlations involved would strengthen the relevance and clarity of the model’s external factors.
In the manuscript, the term PM (Particulate Matter) is used as one of the air pollution indicators. However, it is not clearly specified which specific PM type is utilized in the study (e.g., PM10, PM2.5, or PM1). Since different PM sizes have varying environmental and health implications, and may affect traffic-related predictions differently, it would be helpful if the authors could clarify which PM metric was employed in the analysis and provide a brief justification for its selection.
In Figure 6, 7, 8, 9, and 10 the term "true" is used to represent the actual values in the prediction comparison plot. For clarity and consistency with standard terminology in regression and forecasting tasks, I suggest replacing "true" with "actual." This minor revision would improve the interpretability of the figure for a broader audience.
The literature review provides a useful background for the study; however, many of the cited references appear to be relatively outdated. To enhance the relevance and credibility of the manuscript, I strongly recommend updating the literature with more recent and up-to-date studies, particularly from the last 1-2 years. This will help to better position the proposed method within the current state of research and demonstrate its novelty more clearly.
Recommendation: Based on the aforementioned concerns, I regret to recommend rejection of the manuscript in its current form.
Reviewer 3 Report
Comments and Suggestions for Authors
Thanks to the authors for making the necessary changes to the article. However, a minor revision to the English version is necessary.
Comments on the Quality of English LanguageThanks to the authors for making the necessary changes to the article. However, a minor revision to the English version is necessary.
Reviewer 4 Report
Comments and Suggestions for Authors
The paper proposes a novel GCN-GRU hybrid model integrating spatial heterogeneity and air quality data for traffic flow prediction. The approach addresses non-linear spatio-temporal correlations beyond road adjacency, which is a significant contribution. While the methodology is innovative and experiments demonstrate clear improvements over baselines, the paper requires major revisions for clarity, technical depth, and reproducibility.
1.IN Sec 3.3.1, the derivation of aggregation weights P_n^t
lacks justification. Why is a vector? How does this capture temporal patterns? Clarify the physical interpretation of u_n (Eq 4).2. In Sec 3.4.2 the clustering similarity metric
(Eq 11) is misapplied. evaluates clustering alignment, not traffic-pollutant causality.3. Air quality data (Sec 4.1) is described as dynamic (100×4320 matrix), but preprocessing steps (normalization, handling missing values) are omitted.
4. The "CO+Enhanced" model’s gains (4.43–5.17% RMSE reduction) are significant, but the isolated impact of topology enhancement vs. air quality is conflated. Please disentangle these effects.
5. paarameters for the TC encoder (Eq 2), GCN layers (Sec 3.5), and GRU units (Eq 18–21) are undefined (e.g., filter sizes, layer counts).
6. Author shold avoid hyperboles (e.g., "notable improvement," "significantly enhances")a and use quantitative statements.
7. More relavant references are missing, the rekated work part shoiuld be ehanced.
Round 2
Reviewer 2 Report
Comments and Suggestions for Authors
I have had the opportunity to review the revised version of the manuscript titled "Spatio-Temporal Heterogeneity-Oriented Graph Convolutional Network for Traffic Flow Prediction." However, the revisions and responses provided by the authors have not changed my opinion that the use of outdated data significantly undermines the scientific merit of the study.
From a reviewer's and reader's perspective, the use of data from 2014 forms the core foundation of the study. Therefore, I continue to insist on the necessity of using more recent datasets. I respectfully disagree with the authors' claim that "there are very few datasets that offer both high-resolution traffic flow (5-minute intervals) and temporally and spatially aligned air quality measurements." I believe that in many countries today, due to advances in sensing technologies and the proliferation of monitoring stations, it is indeed possible to access high-resolution traffic and environmental data. Since 2014, significant developments in intelligent transportation systems have enabled the implementation of dynamic intersections and the widespread use of AI-based camera systems, making cities increasingly smart. Furthermore, in alignment with Sustainable Development Goals (SDGs) and national policies, relevant authorities have taken serious environmental measures, which have led to the expansion of air quality monitoring infrastructure in urban areas. As a result, I am convinced that datasets with the characteristics required by this study are accessible today.
Therefore, I still believe that this manuscript is not suitable for publication.
Regards
Reviewer 4 Report
Comments and Suggestions for Authors
The authors have well addressed all my concerns. This paper can be accepted now.
Author Response
We appreciate the reviewer's decision to accept our manuscript.